# Frequency Analysis of Hydrological Data for Urban Floods—Review of Traditional Methods and Recent Developments, Especially an Introduction of Japanese Proper Methods

Chiharu Mizuki [1,2,3,*] and Yasuhisa Kuzuha [1,2]

1   Disaster Mitigation Research Center, Mie University, 1577 Kurima-Machiya, Tsu 514-8507, Japan; kuzuha@crc.mie-u.ac.jp
2   Mie Disaster Mitigation Center, Mie Prefecture and Mie University, 1577 Kurima-Machiya, Tsu 514-8507, Japan
3   Graduate School of Regional Innovation Studies, Mie University, 1577 Kurima-Machiya, Tsu 514-8507, Japan
*   Correspondence: mizkichi@crc.mie-u.ac.jp

**Abstract:** Frequency analysis has long been an important theme of hydrology research. Although meteorological techniques (physical approaches) such as radar nowcasting, remote sensing, and forecasting heavy rainfall events using meteorological simulation models are quite effective for urban disaster prevention, statistical and stochastic theories that include frequency analysis, which are usually used in flood control plans, are also valuable for flood control plans for disaster prevention. Master plans for flood control projects in urban areas often use the concept of T-year hydrological values with a T-year return period. A flood control target is a "landside area that is safe against heavy rainfall or floods with a return period of T years". This review emphasizes discussions of parameter estimation of stochastic models and selection of optimal statistical models, which include evaluation of goodness-of-fit techniques of statistical models. Based on those results, the authors criticize Japanese standard procedures recommended by the central government. Consistency between parameter estimation and evaluation of goodness-of-fit is necessary. From this perspective, we recommend using the maximum likelihood method and AIC, both of which are related to Kullback–Leibler divergence. If one prefers using SLSC, we recommend not SLSC itself but SLSC's non-exceedance probability. One important purpose of this review is the introduction of well-used Japanese methods. Because some techniques that are slightly different from the international standard have been used for many years in Japan, we introduce those in the review article.

**Keywords:** AIC; frequency analysis; goodness-of-fit; maximum likelihood method; parameter estimation; stochastic model; statistical test

## 1. Introduction

First, we would like to emphasize that parameter-estimation processes and processes for selecting the optimal probability distribution are the most important processes in hydrological frequency analyses. Therefore, we focus on only these techniques in this review article. Nevertheless, we have used similar techniques for the past three or four decades. We cannot propose optimal and decisive techniques that numerous researchers think are the optimal techniques.

For preventing water-related disasters, flood control plans are usually made for large rivers. In Japan, main rivers are designated as "Class-1 rivers" in principle, managed by the central government, or as "Class-2 rivers" managed by local governments. A certain numerical goal is set in a flood control plan, for which the jurisdictional government has a responsibility to protect people, residences, and other properties in the river basin.

According to Nakamura [1], such numerical goals are set using two methods. Japan, the Netherlands, the Philippines, and other countries have adopted stochastic goals: T-year

hydrological values with a return period of T years. The United States, China, and other nations have adopted historical maximum values. This review specifically examines the former case. For the former case, the government estimates T-year hydrological values. The estimation processes are divisible mainly into methods of two kinds: non-parametric and parametric methods. Takara [2] described that non-parametric methods can be adopted when the sample size is sufficiently large.

Non-parametric methods are likely to be superior to parametric methods because they use no specific probability distribution: neither a parameter nor an optimal probability distribution need to be selected. By contrast, using the parametric methods, one must estimate parameters and select the optimal probability distribution. Selecting parameters and probability distribution processes include subjective judgments. If one uses parametric methods, subjective judgment must be eliminated to the greatest degree possible. The "Japanese MLIT (Ministry of Land, Infrastructure, Transport and Tourism) flow chart" described later includes some subjective judgment. Therefore, the authors are critical of the method. One should not refer to the flow chart by Japanese MLIT.

The next chapter briefly presents international standard procedures used for hydrological frequency analysis. Because some techniques used in Japan are slightly different from the international standard, we introduce those in Section 3. Techniques described in Sections 2 and 3 are those which have been used for many years. Section 4 presents other techniques developed in recent years. Subsequently, we introduce some future perspectives.

## 2. International Standard Procedure

The World Meteorological Organization (WMO) published its "Guide to Hydrological Practices (WMO-No. 168 fifth edition)" [3] in 1994. One chapter has the title "Frequency analysis (Chapter 27)". The chapter includes the statement that "hydrological phenomena that are commonly described by frequency analysis are storm precipitation and annual flood maxima". They present 16 probability distributions that are commonly used in hydrology. These include a lognormal distribution, Pearson type three distribution, Gumbel distribution, general extreme value distribution, and others, which have been used for hydrologically extreme values. The sixth edition of the guide [4] was published later, and the Kolmogorov–Smirnov test, the probability plot correlation test, AIC, and BIC were introduced. Those are related to the goodness-of-fit test. Moreover, the L-moment method was also mentioned in the guide.

In the "Handbook of Hydrology" [5], one chapter has the title "Frequency Analysis of Extreme Events". As a parameter estimation method, the authors first introduced the method of moments (MOM), the method of L-moments, and maximum likelihood. They describe that maximum likelihood estimators (MLEs) have very good statistical properties for large samples. Experience has shown that they generally perform well with data from records available from hydrology studies, but experience has also shown that MLEs often cannot be reduced to simple formulas. Regarding the selection of the optimal probability distribution, the authors described goodness-of-fit tests and L-moment diagrams. The textbook introduces the Kolmogorov–Smirnov test, the probability plot correlation coefficient test, L-moment diagrams [6], and ratio tests.

Rao and Hamed [7] explicitly described the selection of distributions. After reviewing many reports of the literature, including reports by Hazen [8], Markovic [9], Gupta [10], McCuen and Rawls [11], McCuen [12], Campbell and Sidel [13], Turkman [14], Vogel [15], Vogel and McMartin [16], Haktanir [17], Bobee et al. [18], and Onoz and Bayazit [19], they expounded the chi-square test, Kolmogorov–Smirnoff test, and Akaike's Information Criterion (AIC) [20]. Using these three methods, they described that probability distributions for flood frequency analysis had been selected.

## 3. Japanese History of Estimating T-Year Hydrological Value

As described in Section 2, Akaike proposed the information criterion—AIC [20]. Moreover, many researchers developed their own statistical hydrological theories. We have

an impression that some hydrological procedures used in Japan differ somewhat from international standard procedures. Some effective theories might not be known worldwide because they have been published only in Japanese-language journals.

In Japan, the main class-1 rivers are managed by MLIT. An organization related to MLIT published some manuals [21–23] in which they explained river plan production.

### 3.1. Iwai Method for Parameter Estimation of a Three-Parameter Lognormal Distribution

Iwai [24] proposed their method, which belongs to the "quantile method" type and is used for parameter estimation of the three parameters lognormal distribution. The so-called "Slade type [25] of lognormal distribution" has a bounded probability distribution function. Iwai used "Slade type II" described below. His method consists of the estimation of parameters of the three-parameter lognormal distribution. First, one can define a cumulative distribution function $F(x)$ as explained below. In Equation (1), $\xi$ is designated as "reduced variate" (Equation (2)).

$$F(x) = \frac{1}{\sqrt{\pi}} \int_{-\infty}^{\xi} \exp\left(-t^2\right) dt \tag{1}$$

$$\xi = \alpha \log_{10} \frac{x + b}{x_0 + b} \tag{2}$$

This function for the lognormal distribution has three parameters: $\alpha$, $x_0$, and b. Additionally, $-b$ is a lower bound ($x > -b$). After Kadoya [26] proposed a modification of the original Iwai method, the modified Iwai method has come to be used in most cases. Therefore, we intend to present the "modified Iwai method" herein.

Presuming that there are extreme data with sample size n, then they are annual maxima data. We present these samples as $x_n$ (i = 1, 2, 3, ... , n), which is the ascending order of statistics.

A.    Approximation of $x_0$

First, we use Equation (3) to estimate $x_g$ (approximation of $x_0$)

$$\log_{10} x_g = \frac{\sum\limits_{i=1}^{n} \log_{10} x_i}{n} \tag{3}$$

B.    Estimation of b and $x_0$

First, we produce $b_{(s)i}$ ($i = 1, 2, 3, \ldots m$) values (Equation (4)). Integer m is the nearest integer to $n/10$.

$$b_{(s)i} = \frac{x_i x_{n-i+1} - x_g{}^2}{2x_g - (x_i + x_{n-i+1})} \tag{4}$$

Then, $b$ is estimated using the following equation.

$$\widehat{b} = \frac{1}{m} \sum_{i=1}^{m} b_{(s)i} \tag{5}$$

By defining $X_i = \log_{10}(x_i + b)$, $\widehat{x_0}$ can be estimated by solving the following Equation (6).

$$\log_{10}(\widehat{x_0} + b) = \frac{1}{n} \sum_{i=1}^{n} \log_{10}(x_i + b) = \frac{1}{n} \sum_{i=1}^{n} X_i \tag{6}$$

In this Equation, $\widehat{x_0}$ is the estimate of $x_0$; $\widehat{b}$ is obtained using Equation (5), and is substituted for $b$ in Equation (6).

C.    Final process: Estimation of $\alpha$.

$\alpha$ is estimated by solving the following equation.

$$\frac{1}{\hat{\alpha}} = \sqrt{\frac{2n}{n-1}} \sqrt{\overline{X^2} - (\overline{X})^2} \tag{7}$$

In Equation (7), $\overline{X} = \frac{1}{n}\sum\limits_{i=1}^{n} X_i$ and $\overline{X^2} = \frac{1}{n}\sum\limits_{i=1}^{n} X_i{}^2$.

### 3.2. Ishihara–Takase Method for Parameter Estimation of Three-Parameter Lognormal Distribution

Ishihara and Takase proposed their method [27], which belongs to the "moment method" type. Their method, similar to the Iwai method, estimates the parameters of the three-parameter lognormal distribution. Although a natural logarithm can be used instead of a common logarithm, we use Equation (2) for assigning priority to uniformity with the Iwai method described above.

First, we calculate the sample average $\overline{x}$, standard deviation s, and coefficient of skewness $C_{S1}$. These are estimated using Equations (8)–(10) presented below.

$$\overline{x} = \frac{1}{n}\sum_{i=1}^{n} x_i \tag{8}$$

$$s^2 = \frac{1}{n-1}\sum_{i=1}^{n}(x_i - \overline{x})^2 = \frac{n}{n-1}\left(\overline{x^2} - (\overline{x})^2\right), s = \sqrt{s^2} \tag{9}$$

$$C_{S1} = \sum_{i=1}^{n}(x_i - \overline{x})^3/s^3/(n-1) \tag{10}$$

The parameters that must be estimated are $\alpha$, b, and $x_0$. Ishihara and Takase concluded that $\alpha$ is estimated using the following Equation (11).

$$k = 1/\left(\sqrt{2}\sqrt{\ln\left[-1 + \frac{2^{1/3}}{\left(2+C_S{}^2+\sqrt{4C_S{}^2+C_S{}^4}\right)^{1/3}} + \frac{\left(2+C_S{}^2+\sqrt{4C_S{}^2+C_S{}^4}\right)^{1/3}}{2^{1/3}}\right]}\right)$$
$$\alpha = k \cdot \ln 10 \tag{11}$$

The reason for using $k$ is that their original paper adopted natural instead of common logarithms in Equation (2); $k$ is a parameter for the case of using natural logarithms. Furthermore, $C_S$ is not $C_{S1}$ in Equation (10). Therein, $C_{S1}$ is biased; $C_S$ is corrected when using correction factor $F_{C_S}$ in Equation (12).

$$C_S = C_{S1}(1 + F_{C_S}) \tag{12}$$

As for the correction factor $F_{C_S}$, Ishihara and Takase showed it using a figure. One can obtain $F_{C_S}$, which is a function of sample size n and $C_{S1}$, using their figure, which is well-known as Ishihara–Takase's figure. However, calculating $F_{C_S}$ by PC can be performed more easily than ever using the following procedure. Therefore, we recommend that analysts calculate $F_{C_S}$ by themselves.

A.    Estimating tentative $k$ and $\alpha$ using Equation (11)

First, we estimate $k$ and $\alpha$. In Equation (11), $C_{S1}$ is substituted into $C_S$. $C_{S1}$ is calculated using Equation (10); sample $x_i$ represents observed data.

B.    Generating $\xi_i$ $(i = 1, 2, 3, \dots . n)$

According to Hazen's plotting position formula (as for plotting position formula, see [8,28,29]), $F_i(i = 1, 2, 3, \dots . n$, which is the probability of non-exceedance) is calculated.

Additionally, $\xi_i$ is calculated by the inverse function of Equation (1) as $\xi(F)$. Hazen's plotting position formula is the following, where $i$ is the order of ascending-order statistics:

$$F_i = \frac{2i - 1}{2n} \tag{13}$$

The method for obtaining $\xi(F)$ using the inverse function depends on the software used. Equation (1) can be written as $F(\xi) = \{1 + Erf(\xi)\}/2$, where $Erf(\cdot)$ is the error function. Therefore, an inverse function of it can be expressed as the following Equation (14).

$$\xi(F) = Erf^{-1}(2F - 1) \tag{14}$$

$y_i$ is obtained using an inverse function of Equation (2): —Equation (15). Then we can use $x_0 = 1$, $b = 0$ for simplicity of calculation.

$$\begin{aligned} y(\xi) &= 10^{\xi/\alpha}(x_0 + b) - b \\ &= e^{\xi/k}(x_0 + b) - b \end{aligned} \tag{15}$$

C.    Calculating $C^*{}_{S1\_y}$ and $C^*{}_{S\_y}$ of samples

We can calculate $C^*{}_{S1\_y}$ using $y_i$, which is the coefficient of skewness not of $x_i$ but of $y_i$ by Equation (10). Then, we obtain the theoretical coefficient of skewness $C^*{}_{S\_y}$ using the following Equation (16). When one calculates $C^*{}_{S\_y}$, $k$ is the value estimated by using sample $x_i$ first.

$$C^*{}_{S\_y} = \frac{\exp\left(9/(4k^2)\right) - 3\exp\left(5/(4k^2)\right) + 2\exp\left(3/(4k^2)\right)}{\left(\exp(1/(k^2)) - \exp(1/(2k^2))\right)^{3/2}} \tag{16}$$

As a result, $F_{C_S} = C^*{}_{S\_y}/C^*{}_{S1\_y} - 1$ is obtained.

D.    Calculating three parameters

Using the corrected coefficient of skewness, $k$ or $\alpha$ is obtained. Then $b$ and $x_0$ are estimated using the following equations (Iwai and Ishiguro [30]).

$$\begin{aligned} \lambda &= \exp\left(1/(4k^2)\right) \\ b &= \frac{1}{\sqrt{\lambda^2 - 1}}\sigma - \overline{x} \\ x_0 &= \overline{x} - \frac{\lambda - 1}{\lambda\sqrt{\lambda^2 - 1}}\sigma \end{aligned} \tag{17}$$

In Equation (17), $\overline{x}$ and $\sigma$, respectively, denote the average and standard deviation of the observed sample $x_i$.

*3.3. Etoh's Distribution*

Etoh et al. proposed the probability distribution for extreme values (Etoh et al. [31]). The cumulative distribution function of Etoh's distribution, which has the two parameters a and b, is the following. This probability density function has a heavy tail.

$$F(x) = \begin{cases} \exp\left\{-a\left(1 + \sqrt{bx}\right)\exp\left(-\sqrt{bx}\right)\right\} & (x \geq 0) \\ 0 & (x < 0) \end{cases} \tag{18}$$

Although the following probability density function (Equation (19)) has been used, $F(0)$ (left-hand limit) is $e^{-a}$. It is not zero in accordance with Equation (18). Therefore,

Hayashi et al. [32] proposed the modified function as Equation (20), where $\delta(x)$ represents Dirac's delta.

$$f(x) = \frac{ab}{2} \exp\left\{-\sqrt{bx} - a\left(1 + \sqrt{bx}\right)\exp\left(-\sqrt{bx}\right)\right\} \ (x \geq 0) \tag{19}$$

$$f(x) = \begin{cases} \frac{ab}{2} \exp\left\{-\sqrt{bx} - a\left(1 + \sqrt{bx}\right)\exp\left(-\sqrt{bx}\right)\right\} + \delta(x)\exp(-a) \ (x \geq 0) \\ 0 \qquad\qquad\qquad\qquad\qquad\qquad\qquad\qquad\qquad\qquad\qquad (x < 0) \end{cases} \tag{20}$$

Because $\exp(-a)$ is usually small, however, the use of Equation (19) is adequate.

As a parameter estimation method, we usually use the maximum likelihood method. Etoh et al. [31] and Hoshi [33] recommend the following procedure. The log-likelihood of this probability distribution is presented as Equation (21).

$$\begin{aligned} L(a,b) &= \sum_{j=1}^{N} \ln f\left(x_j\right) \\ &= N\ln a + N\ln b - N\ln 2 - \sum_{j=1}^{N} \sqrt{bx_j} \\ &\quad -a\left[\sum_{j=1}^{N}\exp\left(-\sqrt{bx_j}\right) + \sum_{j=1}^{N}\sqrt{bx_j}\exp\left(-\sqrt{bx_j}\right)\right] \end{aligned} \tag{21}$$

By solving $\frac{\partial L}{\partial b} = 0$, we can obtain $a$, which is a function of $b$, as the following Equation (22), which is referred to as $a_1$.

$$a_1 = \frac{\sum_{j=1}^{N}\sqrt{bx_j} - 2N}{\sum_{j=1}^{N} bx_j\exp\left(-\sqrt{bx_j}\right)} \tag{22}$$

Then, substituting $a_1$, obtained by Equation (22) into Equation (21), $L(a,b)$ is modified to $L(b)$. Finally, we seek the largest $L(b)$—the optimal $b$ in some way. Kubota [34] proposed the following procedure. Solving $\frac{\partial L}{\partial a} = 0$, one can obtain $a$ (designated as "$a_2$"), which is a function of $b$, from Equation (23).

$$a_2 = \frac{N}{\sum_{j=1}^{N}\exp\left(-\sqrt{bx_j}\right) + \sum_{j=1}^{N}\sqrt{bx_j}\exp\left(-\sqrt{bx_j}\right)} \tag{23}$$

The solution of $a$ can be obtained by minimizing $h(b) = |a_1 - a_2|$ [33], which can be performed easily using software such as Mathematica [34]. In Japan, Etoh's distribution is thought to be appropriate for extreme values data. Kuzuha and Mizuki [35] applied several probability distributions to 42,500 pieces of annual maximum one-hour rainfall data whose sample size is 60. They reported that Etoh's distribution was most appropriate for 37% of the 42,500 data. Two-parameters lognormal distribution was most appropriate for 42%, and the Gumbel distribution was most appropriate for 14%.

### 3.4. Approach Proposed by Tsuchiya and Takeuchi

Although Etoh's distribution is quite an effective probability distribution, the L-moment solution has not been known. This probability distribution was not described by Hosking and Wallis [6] because this probability distribution is not well-known internationally.

Tsuchiya et al. [36] (see also Kuzuha [37]) presented the PWM solution of this probability distribution as follows. Their solution was obtained using numerical procedures, but the method is simple.

Specifically, we can estimate the parameters using the following procedure.

$$\beta_r = \int_0^1 x(F)F^r dF = \int_0^\infty x(F)F^r f(x)dx \tag{24}$$

$$
\begin{aligned}
M_{1,0,0} &= \beta_0 = \int_0^\infty x f(x)dx = \frac{ab}{2}\int_0^\infty x \exp\left\{-\sqrt{bx} - a\left(1+\sqrt{bx}\right)\exp\left(-\sqrt{bx}\right)\right\}dx \\
M_{1,1,0} &= \beta_1 = \int_0^\infty x F(x) f(x)dx \\
&= \frac{ab}{2}\int_0^\infty x \exp\left\{-\sqrt{bx} - 2a\left(1+\sqrt{bx}\right)\exp\left(-\sqrt{bx}\right)\right\}dx
\end{aligned} \tag{25}
$$

Equations (24) and (25) indicate the first-order and second-order probability weighted moments. Equation (26) presents the sample probability weighted moments.

$$
\begin{aligned}
\widehat{M}_{1,0,0} &= b_0 = \frac{1}{n}\sum_{i=1}^n x_i \\
\widehat{M}_{1,1,0} &= b_1 = \frac{1}{n}\sum_{i=1}^n x_i\frac{i-1}{n-1}
\end{aligned} \tag{26}
$$

As Tsuchiya and Takeuchi reported [36], $M_{1,1,0}/M_{1,0,0}$ is independent of $b$; it is a function of only $a$. Therefore, we can ignore $b$ and can set $b = 1$ as the following Equation (27).

$$\frac{\int_0^\infty x \exp\left\{-\sqrt{x} - 2a\left(1+\sqrt{x}\right)\exp\left(-\sqrt{x}\right)\right\}dx}{\int_0^\infty x \exp\left\{-\sqrt{x} - a\left(1+\sqrt{x}\right)\exp\left(-\sqrt{x}\right)\right\}dx} = \frac{b_1}{b_0} \tag{27}$$

$a$ is obtained by numerically solving Equation (27).

Finally, estimation of $b$ is obtained by numerical solution of Equation (28) after substituting the $\widehat{a}$ obtained into $a$.

$$\widehat{M}_{1,0,0} = \frac{ab}{2}\int_0^\infty x \exp\left\{-\sqrt{bx} - a\left(1+\sqrt{bx}\right)\exp\left(-\sqrt{bx}\right)\right\}dx \tag{28}$$

Furthermore, we would like to mention the following facts. Takeuchi and Tsuchiya reported the PWM solution of the normal distribution [38], a lognormal distribution, and Pearson type three distribution [39]. Because their findings were published in a Japanese journal, they have not become well-known internationally, but they found their solution ahead of the international hydrological community.

### 3.5. Ueda–Kawamura's Criterion for Evaluating Goodness-of-Fit

Ueda and Kawamura [40] proposed a criterion to evaluate the goodness-of-fit of a probability model. Although many textbooks have recommended the evaluation of the validity of a probability model based on probability studies, it is difficult to evaluate their validity quantitatively. They sought to quantitively evaluate the probability model's goodness-of-fit, as explained below.

A.　Presuming sample data with size $n$ and that have ascending order statistics, then using the plotting position formula, the non-exceedance probability $F_P(x_i)$ is estimated. Several plotting position formulas are expressed as Equation (29).

$$F_P(x_i) = \frac{i-\alpha}{n+1-2\alpha-\beta} \tag{29}$$

For example, for Cunnan's formula [29], $\alpha$ is 0.4 and $\beta$ is 0.

B.　If the cumulative distribution function of the probability model is $F(x)$, then, of course, the non-exceedance probability is $F(x_i)$.

C. Ueda and Kawamura plot $(F(x_i), F_P(x_i))$ on a graph with the normal axis. The minimum and maximum of both axes are 0 and 1. From the viewpoint of goodness-of-fit, the data shown are near the line of $y = x$.

Ueda and Kawamura proposed the use of the $\chi^2$ test as a goodness-of-fit test. As a result, the $\chi^2$ value of each probability distribution is a candidate "fair criterion" when choosing a probability distribution.

### 3.6. Takasao–Takara's SLSC for Evaluating Goodness-of-Fit

Takasao et al. [41] proposed the standard least-squares criterion for goodness of fit (SLSC). This criterion evaluates goodness-of-fit by linearity on a probability plotting paper. The SLSC is expressed as the following Equation (30).

$$\text{SLSC} = \frac{\sqrt{\sum_{i=1}^{n} (s_i - s^*_i)^2 / n}}{|s_{0.99} - s_{0.01}|} \tag{30}$$

In Equation (30), $s$ is a reduced variate and calculated according to Equation (31), where $\xi$ and $\alpha$, respectively, denote the location and scale parameter.

$$s = (x - \xi) / \alpha \tag{31}$$

$$x_i^* = x(F(x_i)) \tag{32}$$

The value of $x_i^*$ is calculated using Equation (32); in addition, $s^*_i$ is transformed from $x_i^*$ by Equation (31). One can assume a probability plotting paper with a horizontal axis $x$ and vertical axis $s$. $(x_i, s_i)$ is on a linear line because of the definition. However, $(x_i, s^*_i)$ is plotted nearly as a straight line but not on the line: SLSC is the mean distance between a straight line and $(x_i, s^*_i)$. That is, SLSC evaluates the mean distance which is the degree of separation of the probability model from the sample, not by vision but by values.

Takasao et al. used a denominator of the right side of Equation (30) to maintain the fairness of the criterion. They regarded vertical scales of the probability plotting paper of each probability distribution as corrected to the same scale, divided by the denominator. As Kuzuha [42] and Hayashi et al. [43] found and Kuzuha et al. [35] [44–46] later examined in detail, however, it is not true. That point is explained in the next section.

### 3.7. Procedure for Parameter Estimation and Choosing the Probability Distribution of the Japan Ministry of Land, Infrastructure, Transport, and Tourism

For estimating long-term stochastic hydrological values (e.g., 100-year precipitation whose return period is 100 years), THE MLIT used their own flow chart for parameter estimation and for choosing an optimal probability distribution [21]. Recently, Kuzuha and Mizuki criticized the flow chart. The flow chart has several shortcomings, but it is wholly inappropriate for three main reasons:

(1) The most important process of the MLIT flow chart is THE evaluation of goodness-of-fit by SLSC and the evaluation of variability by resampling technique for each probability distribution: some candidates are first chosen for the optimal probability distribution by SLSC. However, the authors found that SLSC is not valid from the perspective of fairness among probability distributions. An unfair referee should not judge the match.

(2) In the MLIT flow chart, the probability distribution with the smallest variability is thought to be the optimal one among the candidates selected above. They regard the probability distribution having the least variability of T-year values as optimal. They use three criteria in the flow chart: "at parameter-estimation process", "at process of selecting the optimal", and "at evaluating variability".

(3) The criteria of the least likelihood method for parameter estimation and AIC are related to Kullback–Leibler divergence [47]. If they use the L-moment method (or the

"conventional" moment method), they assign importance to the coincidence of the L-moment (moment) between the model and data.

(4) Work by Tanaka and Takara [48] probably affected the MLIT flow chart the most. Tanaka and Takara mentioned that "if SLSC is less than 0.04, we regard that the probability distribution's goodness-of-fit as sufficient. If using 0.03 for the threshold, most probability distributions are evaluated as inappropriate from the viewpoint of the goodness-of-fit. Then, we use 0.04 as the threshold". The authors have criticized this rationale as it is not scientific. It is for the convenience of administration—the Japan MLIT.

*3.8. Current Best Practice*

We think that consistency is extremely important between the processes of parameter estimation and choosing the probability distribution. In this context, "consistency" means using the same or similar criterion for parameter-estimation and evaluation of goodness-of-fit. Moreover, we believe that "evaluating variability in MLIT flow chart" is not necessary. Let us explain the reason in detail. The most important is that the criterion for evaluating the goodness-of-fit is a fair criterion from the perspective of comparing probability distributions. Because we compare a goodness-of-fit-measure of each probability distribution and select the optimal probability distribution, fairness is most important. From this perspective, SLSC is not a fair measure at all.

Suppose that an analyst uses the maximum likelihood method for parameter estimation and that they estimate parameters of an A-probability distribution and a B-probability distribution. Moreover, suppose that the analyst chooses Takasao–Takara's criterion (SLSC) for selecting the optimal probability distribution. Parameters are selected to maximize the likelihood. Then, the A-probability distribution and B-probability distribution are compared. If the SLSC of the A-probability distribution is smaller than that of the B-probability distribution, the A-probability is selected as the optimal distribution. This poses a big problem since there is a possibility that other parameter sets are selected, and the B-probability distribution is selected as the optimal distribution if parameters are selected to minimize SLSC. This is the reason why we insist that the consistency of measure for parameter-estimation and evaluating goodness-of-fit is quite important.

According to the arguments presented above, using the maximum likelihood method for parameter estimation and using AIC for testing goodness-of-fit are recommended procedures. The main reason is that both are related to Kullback–Leibler divergence [46]. As described in Section 3.7, Tanaka and Takara's explanation [47] for the threshold (=0.04) is inappropriate. However, one can understand the difficulty of policymakers in government agencies in changing their methods quickly to align with an academic perspective. Therefore, we presented some issues related to the conventional method in earlier reports [35,44,45].

A. We recommend using the maximum likelihood method and AIC (or TIC, etc.).
B. If an analyst prefers using SLSC, then we recommend not using SLSC itself but SLSC's non-exceedance probability $F(SLSC)$. For calculating $F(SLSC)$, one must know SLSC's probability distribution function. Hayashi et al. [43] and Kuzuha and Mizuki [35,44] demonstrated how to obtain the SLSC's probability density function using Monte Carlo simulation.
C. If an analyst uses the SLSC's non-exceedance probability, then they can evaluate the goodness-of-fit of each probability distribution, even if SLSC is not a fair criterion. That procedure can be applied to any criterion, even if the criterion is not a fair one from the viewpoint of comparing the degrees of goodness-of-fit.

## 4. Novel Techniques and Future Perspectives

In 2004, Gelder [49] described some well-known techniques for parameter estimation: the method of moments (MOM), maximum likelihood estimation (MLE), least squares, Bayesian estimation, minimum cross-entropy, probability weighted moments (PWMs), and

L-moments. More recent reports, such as that by Yuan et al. (2018) [50], described the adoption of the so-called MOM for parameter estimation. Langat et al. (2019) [51] adopted MLE after reviewing some techniques. Those are MOM, L-moments, LH moments [52], and the expected moments algorithm (EMA). Anghel and Ilinca (2022) [53] used both MOM and L-moments for parameter estimation.

Coles [54] and Hayashi et al. [43] considered non-stationary hydrological models. Hayashi et al. discussed non-stationary hydrological frequency models introducing time-dependent parameters. Their report recommended the use of MLE for parameter estimation. Langat et al. commented on the method of Bayesian estimation: "although there are drawbacks of complexity in its implementation in present time, it might become a useful non-stationarity flood frequency analysis model in the future, with advancements in technology".

Yuan et al. (2018) [50] described that "the choice of an appropriate PDF is still one of the major issues in engineering practice because there is no general agreement as to which distribution could be used for the frequency analysis of extreme rainfalls". They adopted the chi-square test for selecting the optimal probability distribution. Langat et al. [51] introduced the Kolmogorov–Smirnov, Anderson–Darling, and Cramer–Von Mises tests in addition to the chi-square test.

Most techniques described above have a long history; quite attractive and novel techniques that have become a new international standard have not been proposed in recent years. Nevertheless, because hydrological frequency analyses that use non-stationary hydrological data have become increasingly important in light of drastic climate change, non-stationary analyses have become ever more necessary. Some techniques are useful for non-stationary analyses. The maximum likelihood method and AIC, TIC, or BIC, which are related to Kullback–Leibler divergence [47], are expected to be crucially important in the research area. In addition, the method of Bayesian estimation might be particularly effective.

## 5. Conclusions

We reviewed statistical hydrological studies, especially those conducted in Japan. Many Japanese government analysts often use procedures developed in Japan, which have been recommended by Japanese MLIT. We criticized the use of those procedures. Some consistency between parameter estimation and evaluation of goodness-of-fit is necessary. From this perspective, we recommend using the maximum likelihood method and AIC, both of which are related to Kullback–Leibler divergence. If one prefers using SLSC, we recommend not SLSC itself but SLSC's non-exceedance probability.

Techniques for parameter estimation and selecting the optimal probability distribution should be discussed from an international viewpoint. Some techniques related to Kullback–Leibler divergence or Bayesian estimation might be candidates for the solution of non-stationary flood frequency analyses.

**Author Contributions:** Conceptualization, C.M. and Y.K.; methodology, C.M. and Y.K.; investigation, C.M. and Y.K.; writing, C.M.; supervision, Y.K.; project administration, Y.K.; funding acquisition, Y.K. All authors have read and agreed to the published version of the manuscript.

**Funding:** This research was partly supported by JSPS Grants-in-Aid for Scientific Research. The grant Number is JP19K04613.

**Institutional Review Board Statement:** Not applicable.

**Informed Consent Statement:** Not applicable.

**Data Availability Statement:** Not applicable.

**Conflicts of Interest:** The authors declare no conflict of interest.

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
