# Peer review of "Frequency Analysis of Hydrological Data for Urban Floods—Review of Traditional Methods and Recent Developments, Especially an Introduction of Japanese Proper Methods"

_water, doi:10.3390/w15132490_

Round 1

Reviewer 1 Report (New Reviewer)

This paper mainly proposed a number of the probability distribution functions applied in the frequency analysis for the urban flood in Japan. Although the detailed concepts of probability distributions and the good-of-fit methods for identifying the best-fit pdf, several questions should be answered and figured out.

1. Most of the references cited in this paper are published in the reports written in Japanese. More journal papers should be cited and discussed in comparison to the frequency methods commonly used in Japan.

2. Apart from the introduction to the concepts concerned, the model validations for the frequency methods mentioned in this paper should be addressed; it is helpful to understand the above frequency methods.

3. The probability distributions introduced in this paper are rarely referred to the hydrological frequency analysis. Their difference from the well-known probability distributions used in the hydrological frequency analysis (e.g., Person Type III, Gumbel, Extreme Value distributions and so on) should be expressed.

4. In general, the parameters of the probability distributions are calibrated using the statistical moments (Product or L moment) without the AIC algorithm. The reason why the AIC algorithm is superior to conventional statistical moment approaches should be explained.   

Author Response

Reviewer 2 Report (New Reviewer)

The manuscript is well written and interesting. Thank you.

Author Response

Thank you.

Reviewer 3 Report (New Reviewer)

clearer and more concise expressions are required in sections that primarily involve the discussion

Round 2

Reviewer 1 Report (New Reviewer)

non

Reviewer 3 Report (New Reviewer)

Questions are well addressed and the representation is better now.

This manuscript is a resubmission of an earlier submission. The following is a list of the peer review reports and author responses from that submission.

Round 1

Reviewer 1 Report

The manuscript, in its present form, contains numerous (technical/structural and grammatical) weaknesses. I found no major novelty in this research work.The used methods are not original, either. Moreover, this manuscript is not well organized, and is poorly written.This manuscript is far from the level of Water Journal.The overall quality does not meet the standard of the journal.

I recommend the manuscript be rejected.

Author Response

................On the other hand, reviewer #1 did not show any valuable comments. He did not mention any comments at all. I suppose that he did not know “review paper” or he did not notice that our manuscript is categorized into review paper. The proof is that he wrote that “the used methods are not original”.  We tried to not show new technique but review many research achievements, because our manuscript is “review paper”. Anyway, he did not any comments which improve the manuscript. Therefore, we could not react at all.

Reviewer 2 Report

1. Innovativeness and novelty are missing in the manuscript

2. What is the length of the data?

3. Any missing data in your study? If yes, then what approaches you have used to find the missing data?

4. Any assumption have you made in your work?

5. Give citations wherever required.

6. The real objective, purpose, and implications are unclear.

 7. A lot of essential technical details on the modeling (e.g., modeling methods) are missing.

8. The dataset used and modeling results achieved in this study may be interesting to local managers, but in-depth interpretation of the results and discussion on engineering and/or management implications are missing.

9. To meet the journal’s standards, a significant amount of work on data analysis, results interpretation and discussion is necessary, and the manuscript needs to be reorganized and rewritten.

10. Refer below mentioned references which are relevant to your work:-

-- Mangukiya, N. K., Mehta, D. J., & Jariwala, R. (2022). Flood frequency analysis and inundation mapping for lower Narmada basin, India. Water Practice & Technology, 17(2), 612-622.

Author Response

........................

Reviewer #2 made some comments. However, we do not think that we have to react those. Firstly, could you please notice to his comment #10. He was kind and he tried to show us valuable comment to improve manuscript, but the level of paper he recommended is too low level. Please read 3.1.1 and 3.1.2 of their paper. Their paper does not include new technique of parameter estimation and selecting process of probability distribution. This is a proof of that he (reviewer #2) is not an expert of research area which our manuscript tries to mention. He gave us some questions. Please see #2. We did not use data. What does he want to say? Please see #4. We review many research achievements but we did not consider on new theory because our manuscript is review.

Reviewer 3 Report

Thank you for considering me to review the entitled manuscript "Frequency Analysis of Hydrological Data for Urban Floods – Review of Traditional Methods and Recent Developments". Although authors made efforts to provide an overview of discussions of parameter estimation of stochastic models and selection of optimal statistical models. Based on those results, the author criticizes Japanese standard procedures recommended by the central government. The manuscript is relatively incomplete and does not cover novel aspects relating to methods, results, and implications. Authors need to include a broad scope of content from recently published works to improve the quality of the review manuscript. Conclusions need to be expanded. 

Author Response

......

As a conclusion, we have made an effort in order to react to valuable comments of reviewers #3 and #4, and we have spent more than 20 days. Shortly speaking, they requested us to mention broad and novel research achievement by many researchers. As for it, we believe that we could have done it.

Reviewer 4 Report

 Congratulations to the authors for the original way of making the material.

The conclusions of the study represent useful and approved practical recommendations.

we recommend the authors to format the material in accordance with the rules of the journal, eg: L92 [21][22][23] replaced by [21-23]

Author Response

....................As a conclusion, we have made an effort in order to react to valuable comments of reviewers #3 and #4, and we have spent more than 20 days. Shortly speaking, they requested us to mention broad and novel research achievement by many researchers. As for it, we believe that we could have done it.

Round 2

Reviewer 2 Report

Revision is appropriate

Reviewer 3 Report

The current manuscript is not suitable for publication in Water Journal. As I said in a previous report, authors need to extensively work on manuscript structure especially highlighting the broad scope of work toward environmental significance. Need to discuss it more with the latest literature. Highlight the key limitations and methodological gaps that future studies can pay attention to while hydrological analysis for Urban Floods assessment.